

# High-resolution modeling of atmospheric dispersion of dense gas using TWODEE-2.1: application to the 1986 Lake Nyos limnic eruption

Arnau Folch[1], Jordi Barcons[1], Tomofumi Kozono[2], and Antonio Costa[3]

[1]Barcelona Supercomputing Center (BSC), Barcelona, Spain
[2]Department of Geophysics, Graduate School of Science, Tohoku University, Japan
[3]Istituto Nazionale di Geofisica e Vulcanologia (INGV), Bologna, Italy

*Correspondence to:* Arnau Folch (afolch@bsc.es)

**Abstract.** Atmospheric dispersal of a gas denser than air can threat the environment and surrounding communities if the terrain and meteorological conditions favor its accumulation in topographic depressions, thereby reaching toxic concentration levels. Numerical modeling of atmospheric gas dispersion constitutes a useful tool for gas hazard assessment studies, essential for planning risk mitigation actions. In complex terrains, microscale winds and local orographic features can have a

strong influence on the gas cloud behavior, potentially leading to inaccurate results if not captured by coarser-scale modeling. We introduce a methodology for microscale wind field characterization based on transfer functions that couple a mesoscale Numerical Weather Prediction Model with a microscale Computational Fluid Dynamics (CFD) model for the atmospheric boundary layer. The resulting time-dependent high-resolution microscale wind field is used as input for a shallow-layer gas dispersal model (TWODEE-2.1) to simulate the time evolution of $CO_2$ gas concentration at different heights above the terrain.

The strategy is applied to review simulations of the 1986 Lake Nyos event in Cameroon, where a huge $CO_2$ cloud released by a limnic eruption spread downslopes from the lake suffocating thousands of people and animals across the Nyos and adjacent secondary valleys. Besides several new features introduced in the new version of the gas dispersal code (TWODEE-2.1), we have also implemented a novel impact criterion based on the percentage of human fatalities depending on $CO_2$ concentration and exposure time. New model results are quantitatively validated using the reported percentage of fatalities at several

locations. The comparison with previous simulations that assumed coarser-scale steady winds and topography illustrates the importance of high-resolution modeling in complex terrains.

## 1  Introduction

The atmospheric dispersion of gases (of natural, accidental or intentional origins) can be very hazardous to life and the environment. In industry, historic examples of tragic accidents include the dioxin release in Seveso (Italy, 1976), the methyl isocyanate

in Bophal (India, 1984) or the petroleum gas explosions in Mexico City (Mexico, 1984), among several others (e.g. Britter, 1989). From the point of view of natural hazards, cold $CO_2$ gas released from natural Earth degassing can be of concern because, being denser than air, $CO_2$ gas clouds hug the ground and spread downslopes governed by local topography and



near-surface winds. Under particular conditions (*e.g.* atmospheric stability), gas can accumulate in topographic depressions reaching concentration levels toxic for humans and animals. For example, many gas manifestations in Central and South Italy, characterized by persistent $CO_2$ emissions, have caused several periodic accidents (Chiodini et al., 2004). Another case of natural origin, much rarer than diffuse emissions but potentially much more hazardous, are limnic eruptions triggered dur-

ing overturning of $CO_2$ rich volcanic lakes (e.g. Zhang, 1996), such as the eruptions occurred at lakes Monoun and Nyos (Cameroon) in 1984 and 1986 respectively.

The most tragic event occurred on 21 August 1986 at Lake Nyos, when a $CO_2$ gas cloud spread across the surrounding valleys suffocating inhabitants and animals. On occasion of the 30th anniversary of this disaster, the Commission on Volcanic Lakes of the International Association of Volcanology and Chemistry of the Earth's Interior (IAVCEI) organized the workshop "30 years

of Lake Nyos disaster" (Yaounde, 14-20 March 2016) to gather scientific experts from several disciplines with the objective of reviewing the groundbreaking research advances in that field and discuss future roadmaps for research and risk assessment and mitigation (Rouwet et al., 2016). One of the outcomes from this workshop was that, despite the successful mitigation measures taken at these two lakes (*e.g.* installation of degassing pipes, relocation of local communities to safer settlements, etc.) have reduced the level of hazard notably, there is no yet a quantitative hazard assessment from volcanic lakes at Cameroon and

elsewhere in Africa (*e.g.* lake Kivu in Democratic Republic of Congo and Rwanda). In this regard, dense gas dispersal simulations and characterization of eruption/gas emission scenarios constitute the backbone of a probabilistic quantitative hazard assessment.

Few studies have been conducted to simulate atmospheric dispersion of dense $CO_2$ clouds from natural Earth degassing (e.g. Pierret et al., 1992; Costa et al., 2005, 2008; Chiodini et al., 2010; Costa and Chiodini, 2015). In particular, Costa and Chiodini

(2015) have recently used the TWODEE-2.0 model (Hankin and Britter, 1999; Folch et al., 2009) to simulate the $CO_2$ cloud from the 21 August 1986 lake Nyos limnic eruption considering various scenarios for cloud volume and eruption duration estimated from previous studies (Evans et al., 1994; Kanari, 1989; Tuttle et al., 1987). For their simulations, Costa and Chiodini (2015) considered a digital terrain elevation model with a resolution of 90m. Results evidenced the strong effect of the topography and local wind field, leading to different gas flow patterns across the different valleys of the complex topography of

the area. Despite some scenarios could reproduce lethal $CO_2$ concentrations in many of the sites where fatalities did actually occur, these simulations where unable to capture the full dispersal pattern given some model limitations including, but not limited to, steady winds and gas emission rates or insufficient accuracy of near surface winds. This later aspect is particularly critical on very complex terrains, where local-scale (tens of m) wind patterns and the resolution of the Digital Elevation Model (DEM) can have a strong influence on model results.

Based on all these previous considerations, the objective of this paper is to address high-resolution (tens of m) numerical modeling of $CO_2$ atmospheric dispersal in complex terrains. To this purpose, we introduce a methodology for local wind field characterization based on transfer functions that couple mesoscale Numerical Weather Prediction (NWP) models with a microscale Reynolds-Averaged Navier-Stokes (RANS) Computational Fluid Dynamics (CFD) model for the atmospheric boundary layer. The resulting high-resolution time-dependent wind field is given as input to the TWODEE shallow-layer gas

dispersal model to simulate the evolution of gas concentration with time at different heights above the terrain. In addition to



coupling with meso/micro 4D meteorological models, the TWODEE-2.0 code has been improved (updated to version 2.1) in order to overcome a few limitations, including the option to easily describe time-dependent gas sources, heterogeneous terrain roughness, and assessment of the impact through a novel probabilistic approach that estimates the percentage of fatalities depending on gas concentration and exposure time (dose) at near-ground levels. The new version of the code is applied to

reconstruct gas dispersion from the 1986 lake Nyos event compared with previous simulations (Costa and Chiodini, 2015), and results are used to illustrate the model gain on the time evolution of the gas source.

In this manuscript, Section 2 overviews the main characteristics of the TWODEE model and the modifications introduced to account for heterogeneous high-resolution 4D wind fields, time-dependent source term, and terrain variable roughness. The new probabilistic impact criterion for TWODEE model validation is also presented. Section 3 summarizes the events occurred

at lake Nyos and surroundings during 21 and 22 August 1986 and the results (and limitations) from previous simulation studies (Costa and Chiodini, 2015). A novel strategy for high-resolution local wind field characterization on very complex terrains is presented on Section 4. The methodology uses the concept of transfer functions to couple the mesoscale Weather Research and Forecasting (WRF) (Skamarock et al., 2008) model with the CFD code Alya (Houzeaux et al., 2009) adapted to atmospheric boundary layer flows (Avila et al., 2013). Section 5 shows the TWODEE-2.1 model results for the 1986 lake Nyos case of study

and discusses how and why the model new features increase the accuracy of simulations. In addition, we also perform a model parametric study varying the source term to constrain its intensity, evolution, and duration. Finally, Section 6 summarizes and discusses the main results and future developments..

## 2    The TWODEE dense gas dispersal model

TWODEE-2 (Hankin and Britter, 1999; Costa et al., 2008; Folch et al., 2009) is a FORTRAN90 code for the atmospheric

dispersal of dense gases based on the shallow layer approach. Under the assumption that $h/L << 1$ (being $h$ the gas cloud depth and $L$ a characteristic length), the 2D shallow layer approach allows a compromise between more realistic but computationally demanding 3D CFD models and simpler 1D integral models. The TWODEE family models build on the depth averaged equations for a gas cloud resulting from mixing a gas of density $\rho_g$ with an ambient fluid (air) of density $\rho_a$ ($\rho_g > \rho_a$). The integration of volume, mass, and momentum balance equations over the mixed cloud depth from the ground to the top of the

cloud yields to (e.g. Hankin and Britter, 1999):

$$\frac{\partial h}{\partial t} + \nabla \cdot (h\overline{\mathbf{u}}) = u_e + u_s \tag{1}$$

$$\frac{\partial (h(\overline{\rho} - \rho_a))}{\partial t} + \nabla \cdot (h(\overline{\rho} - \rho_a)\overline{\mathbf{u}}) = \rho_a u_e + \rho_g u_s \tag{2}$$





$$\frac{\partial\left(h\overline{\rho}\,\overline{\mathbf{u}}\right)}{\partial t} + \nabla\cdot\left(h\overline{\rho}\,\overline{\mathbf{u}}\otimes\overline{\mathbf{u}}\right) + \frac{1}{2}S_1\nabla\left(g(\overline{\rho}-\rho_a)h^2\right)+$$
$$S_1 g(\overline{\rho}-\rho_a)h\nabla e + \frac{1}{2}\overline{\rho}C_D|\overline{\mathbf{u}}|\overline{\mathbf{u}}+$$
$$\mathbf{F} + \kappa\rho_a\frac{D\left(h(\overline{\mathbf{u}}-\mathbf{u}_a)\right)}{Dt} = \rho_a u_e\mathbf{u}_a \tag{3}$$

where $h$ is the cloud depth (defined as the height below which 95% of the buoyancy is located), $\overline{\rho}$ is the depth averaged cloud density, $\overline{\mathbf{u}} = (\overline{u}_x, \overline{u}_y)$ is the depth averaged cloud velocity, $\mathbf{u}_a$ is the ambient fluid (air) velocity vector, $u_e$ is the ambient fluid entrainment velocity modulus, $u_s$ is the gas inflow velocity modulus (source term), $e = e(x,y)$ is the terrain elevation, $C_D$ is a drag coefficient, $\mathbf{F}$ is the turbulent shear stress force (per unit area), and $S_1 \approx 0.5$ and $\kappa$ are semi-empirical parameters. The terms in the momentum equation (3) include the local time derivative, the convective term, the pressure gradient (assumed hydrostatic although the density profile can be non-uniform), the effect of terrain slope, the surface shear stress (depending on the terrain roughness and characterized by the drag coefficient), the force per unit area exerted by turbulent shear stress and, finally, the leading edge terms that account for interaction among dense and ambient fluids. Given closure equations for the drag coefficient $C_D$, shear stress force $\mathbf{F}$ and entrainment velocity $u_e$ (Hankin and Britter, 1999; Folch et al., 2009), the set of equations above can be resolved numerically to obtain cloud height and vertically averaged density and velocity depending on terrain, source term (definition of $u_s$) and ambient fluid velocity (wind field). Although TWODEE is a shallow water model, it can also estimate the vertical density profile from the depth averaged density $\overline{\rho}$ assuming an empirical exponential decay (Hankin and Britter, 1999):

$$\rho(z) = \rho_a + \frac{2}{S_1}(\overline{\rho}-\rho_a)\exp\left(-\frac{2}{S_1}\frac{z}{h}\right) \quad 0\leq z\leq h \tag{4}$$

from which the vertical concentration profile $c(z)$ (expressed in $\mathrm{ppm}$) and the dosage $Do(t,z)$ during a time interval $(0,t)$ can be computed as:

$$c(z) = c_b + (10^6 - c_b)\frac{\rho(z)-\rho_a}{\rho_g-\rho_a} \tag{5}$$

$$Do(t,z) = \int_0^t [c(z)]^n\, dt \tag{6}$$

where $c_b$ is the background concentration (in $\mathrm{ppm}$) and $n$ is the so-called toxicity exponent.

## 2.1 TWODEE-2.1

The TWODEE code version 2.0 (Folch et al., 2009) has been improved so that the upgraded version 2.1 can deal with:

1. Time-dependent heterogeneous wind fields furnished by mesoscale or microscale wind models. The previous code version admitted only a steady homogenous wind profile or heterogeneous wind fields furnished by a meteorological processor (included in the code distribution package) based on the Diagnostic Wind Model (Douglas and Kessler, 1990,



DWM). The DWM model generates a gridded wind field from input data (observations) at a point of the domain by adjusting the domain-scale mean wind for terrain effects and performing a divergence minimization to ensure mass conservation. This supposes an obvious limitation if winds vary (*e.g.* during long-lasting emissions) or in very complex terrains or large domains, where the winds adjusted by DWM can be notably different depending on the location of the point where observations are considered. In contrast, the updated TWODEE code version can read outputs from either the Weather Research and Forecasting (WRF) (Skamarock et al., 2008) mesoscale model (see Section 4.1) or from CFD microscale model simulations. In the case of WRF, a meteorological pre-processor reads the WRF model outputs for a user-defined time interval ($n_t$ WRF output time steps) and performs an interpolation from the WRF native grid (Arakawa staggered C-grid in the horizontal, pressure levels for the vertical) to a series of user-defined z-cuts on a terrain-following regular grid. In the case of microscale simulations, the same meteorological pre-processor reads outputs from the ALYA-CFDWind model (see Section 4.2) and, for each WRF time slice, performs the meso-to-micro downscaling using transfer functions as explained in Section 4.3. This results on $n_t$ downscaled wind fields on the same set of terrain-following z-cuts. Whichever the approach considered for the wind field (*i.e.* mesoscale WRF or WRF downscaled with ALYA-CDFWind), TWODEE-2.1 reads winds at the gridded z-cuts and then interpolates to its computational mesh at a user-defined reference height.

2. Time-dependent source term(s). TWODEE-2.1 can easily handle multiple point or extended time-dependent source terms, consisting on piece-wise constant pulses of arbitrary duration (the previous version had to use the "restart" option).

3. Multiple input file formats for digital elevation models and terrain roughness. In addition, model output has also been enhanced including netCDF and kml output file formats.

4. Impact assessment for dense $CO_2$ gas dispersal. According to the Occupational Safety and Health Administration (OSHA), $CO_2$ is considered safe for exposures up to 8-hours per day at air concentrations below 0.5% (5.000 ppm). However, at higher concentrations/dosage, $CO_2$ causes several adverse health effects when inhaled. Experience shows that $CO_2$ air concentrations of around 5% (50.000 ppm) produce heavy breathing, sweating, quicker pulse, weak narcotic effects and headache. Under these concentrations, the exposure time to avoid the development of adverse health symptoms is of few minutes only (Costa and Chiodini, 2015). For example, a 30-minute exposure to 5% concentration (50.000 ppm) produces intoxication manifested as headaches, dizziness, restlessness, breathlessness, increased heart rate and blood pressure, and visual distortion. At around 10% concentration (100.000 ppm) humans are affected by respiratory distress, impaired hearing, nausea, vomiting, and loss of consciousness in 10-15 min only. Finally $CO_2$ air concentrations >15% (150.000 ppm) are considered lethal causing coma, convulsions, and rapid death. The U.K. Health and Safety Executive (HSE; www.hse.gov.uk) developed an assessment of dangerous toxic substances, including $CO_2$, defining the specified level of toxicity (SLOT) and the significant likelihood of death (SLOD) depending on concentration and duration of exposure. Based on these, we assume a cumulative normal distribution for the percentage of human





fatalities:

$$P(c,d) = \frac{1}{2}\left[1 + \mathrm{erf}\left(\frac{c - \mu}{\sqrt{2}\sigma}\right)\right] \qquad (7)$$

$$\mu = a_0 + \frac{b_0}{(1 + d^{c_0})} \qquad (8)$$

$$\sigma = a_1 + \frac{b_1}{(1 + d^{c_1})} \qquad (9)$$

where $P$ is the probability of death, $c$ is the $CO_2$ concentration (expressed in %), $d$ is the exposure duration (expressed in minutes), and $a_i$ and $b_i$ are empirical constants. After calibration of eq. (7) with the HSE tabulated values (assuming SLOT at 3%), we obtained the following values for the constants: $a_0 = 5.056$, $b_0 = 17.885$, $c_0 = 0.357$, $a_1 = 0.662$, $b_1 = 2.421$, and $c_1 = 0.354$. Results are shown in Figure 1, plotting the percentage of fatalities (probability of death in %) as function of $CO_2$ concentration for different exposure durations. An impact criterion based on these empirical curves was added in TWODEE-2.1 to compute, at each point of the computational domain, the predicted percentage of fatalities at user-defined heights.

## 3 The 1986 Lake Nyos event and previous modeling results

Lake Nyos (Fig. 2) is one of the ∼40 volcanic lakes scattered along the 1600 km-long Cameroon Volcanic Line (e.g. Lockwood and Rubin, 1989). This lake became famous worldwide on Thursday 21 August 1986 after the occurrence of the most tragic limnic eruption ever registered. During few (<5) hours, a huge (0.1-1 km³) $CO_2$ gas cloud (Tuttle et al., 1987; Kanari, 1989; Evans et al., 1994) released during the lake Nyos overturning spread downslopes from the lake (1100 m a.s.l.) filling up the underlying Nyos valley and suffocating around 1700 people and 3000 cattle (e.g. Kling et al., 1986). Evidence from eyewitness reports indicated that the cloud directed primarily W-NW at around 21:00 LT (20:00 UTC) affecting the bottom valleys of Cha and Fang (see Fig. 3), but without causing reported deaths at the later location (Baxter and Kapila, 1989). Following this initial dispersal phase, the gas cloud direction shifted towards NE, probably as a consequence of a sudden wind veer, filling up the Nyos valley down to the Subum village (∼10 km line of sight from the lake), where the largest number of casualties occurred. Deaths in humans and animals (including birds) occurred up to 20 km distance across the main and adjacent secondary valleys (Le Guern et al., 1992).

Le Guern et al. (1992) collected multiple testimonies of survivors that allowed the authors for reconstructing the history of the event, locating where casualties occurred, and constructing a map of the areas impacted by the disaster (see Table 1 and Fig.1 in Le Guern et al. (1992)). It should be stressed that the percentages of fatalities reported by Le Guern et al. (1992) resulted from posterior interviews by anthropologists and are subject to large uncertainties related to translation and interpretation, approximate location of sites, and actual number of casualties. Notwithstanding these limitations, data inferred from eye-witnessing constitutes the only source of information available for indirect model validation given the lack of any wind or gas



concentration measurement at Nyos on that time. At present, the level of risk at lakes Nyos and Monoun has decreased notably after the successful deployment of degassing pipes started in March 2001 and February 2003 respectively. The progressive gas removal resulted in considerable deepening of the level of gas-rich water in a short period of time (Kling et al., 2005; Kusakabe et al., 2008). However, there is still the recognized need to perform a quantitative $CO_2$ hazard assessment for several reasons,

including the possibility of future gas build-up or a breakthrough of the dam build at the northern shore of the lake. Thus, for example, Aka and Yokoyama (2013) estimated that dam break could cause a sudden drop in lake level by 40 m, followed by decompression and inevitably a new gas burst.

Costa and Chiodini (2015) have recently used the TWODEE-2.0 model to simulate four different scenarios (released gas mass ranging from 0.29 to 1.95 Tg, wrongly reported in their table as Gg) using a 90-m resolution DEM. Surface wind data

resulted from applying the DWM mass-consistent pre-processor to a constant wind profile extracted from the closest point of the NCEP/NCAR Reanalysis-1 (Kalnay et al., 1996). As a result of this limitation, none of their simulations was able to capture properly the cloud dispersion pattern, strongly influenced by a sudden wind veering. Figure 4 shows, for illustrative purposes, results for their scenario-II, the one that better reproduced the observations (see Fig.5 in Costa and Chiodini (2015) for other scenario results). In the following sections, these previous simulations are revisited considering a higher DEM accuracy (30-m

instead of 90-m; from ASTER G-DEM, a product of METI and NASA) and high-resolution transient microscale surface winds derived from downscaling the WRF-ARW mesoscale winds with the ALYA-CFDWind model.

## 4 Wind field characterization

### 4.1 Mesoscale wind modeling using WRF-ARW

The Weather Research Forecast (WRF) is a fully compressible, non hydrostatic mesoscale NWP model and atmospheric

simulation system designed to serve both operational forecasting and atmospheric research needs (Skamarock et al., 2008). The model uses finite differences schemes on a staggered horizontal Arakawa C-grid and a terrain-following vertical coordinate system to solve the atmospheric flow. Here, the version 3.4.1 of the dynamical solver Advanced Research WRF (WRF-ARW) was configured with the physical parameterizations and schemes summarized in Table 3.

For the Nyos application, the WRF-ARW simulation starts on Thursday 21 August 1986 at 00:00 UTC lasting 48 h (around

25 18 h are allowed for model spin-up). Initial and 4-times daily boundary conditions driving WRF come from the NCEP/DOE Reanalysis 2. Figure 5 shows the five domains used, consisting of 1 parent grid at 81 km horizontal resolution, covering the African continent, and 4 nested domains at 27, 9, 3, and 1 km horizontal resolutions, all centered arround Lake Nyos. The regional synoptic situation on Thursday 21 August 1986 is illustrated in Figure 6, showing WRF-ARW results for the first nest domain (27 km resolution) at different times. It can be observed how a low pressure (<800 hPa at 2000 m) region and

30 its cyclonic circulation located NE of Cameroon coexists with strong winds both South and North and a region of weak winds over NW Cameroon at around 16:00 UTC. The simulations indicate that this situation lasted for about 6 hours untill 22:00 UTC, when winds over Nyos became stronger and pointed SE. In contrast, near-surface winds followed a very different pattern (Figure 7) reflecting the topography-induced forcing at lower atmospheric levels. However, given the orographic complexity,



the WRF-ARW results at 1 km resolution (inner nest) are still insufficient for driving high-resolution gas transport simulations, indicating the need for an ulterior downscaling.

## 4.2 Microscale wind modeling using ALYA-CFDWind

ALYA (e.g. Houzeaux et al., 2009; Vázquez et al., 2016) is a High Performance Computing (HPC) multi-physics parallel

solver based on a Finite Element Method (FEM) able to run with thousands of processors with an optimal scalability. Within this multi-physics general framework, Avila et al. (2013) implemented a solver for the Atmospheric Boundary Layer (ABL) based on the Reynolds-Averaged Navier-Stokes (RANS) equations and a $\kappa - \epsilon$ turbulence model adapted to atmospheric flows in complex terrains. This model, called ALYA-CFDWind, can handle thermal coupling assuming the Boussinesq approximation although here we constrain to neutral atmospheric stability. The ALYA-CFDWind module, originally developed in the context

of wind energy, includes Coriolis effects, a consistent limitation of the mixing length, a wall law for atmospheric boundary layers (logarithmic profile depending on terrain roughness and wind friction velocity), automatic meshing and generation of boundary conditions for atmospheric boundary layer wind profiles over a flat terrain. In order to have consistent inflow boundary conditions (*i.e.* flat terrain inflow profile) and also to prevent flow recirculation at the outflow boundary, the ALYA-CFDWind computational domain is made of an external flat buffer designed to accommodate the flow, an adjacent transition

zone, and an inner higher-resolution zone with the real topography and where the flow is actually computed.

For the Nyos case, the ALYA-CFDWind computational domain consists of an inner zone of $20 \times 20$ km$^2$ at 50 m horizontal resolution, a transition zone of 15 km and a flat buffer zone of 10 km to accommodate the flow. Along the vertical direction, the structured hexahedral grid extends up to 5 km above the terrain with 64 vertical layers growing geometrically in size from 0.5 m at surface to 250 m at the top of the computational domain. The resulting computational mesh (see Fig. 8) has a total

number of grid points of about 30 million. The Coriolis force was set to that of a latitude of 6° N, and the maximum mixing length calculated depending on the wind at top as in Apsley and Castro (1997). As boundary conditions, we prescribed the wind at top (geostrophic wind) to a constant reference value of $10 \text{ ms}^{-1}$ considering different geostrophic wind directions (sectors) at 15° intervals. One ALYA-CFDWind simulation was performed for each reference direction of the geostrophic wind (*i.e.* 24 different runs are necessary to scan all possible geostrophic wind directions). Assuming self-similarity, these reference runs

can be scaled and interpolated in direction during the meso-to-micro downscaling strategy (see section 4.3) depending on the mesoscale (WRF) time-dependent geostrophic wind direction and intensity.

## 4.3 Meso-to-micro downscaling strategy

Mesoscale NWP models like WRF-ARW can be used to forecast winds at horizontal resolutions down to around 1 km. This grid resolution may be insufficient to drive subsequent gas dispersion simulations over complex terrains, where sub-grid scale

topographic features can alter near-surface winds and therefore the resulting gas dispersal pattern. For this reason, a meso-to-micro downscaling strategy may be necessary in order to capture wind-forcing effects caused by the local sub-grid topography. At present, model downscaling is being a subject of active research within the atmospheric and wind engineering communities, with two well-differentiated strategies dominating the scene. On the one hand, statistical downscaling methodologies (e.g.



Ranaboldo et al., 2013; Devis et al., 2013; Kirchmeier et al., 2014) build on finding correlations between global/regional model simulations and observations during long periods of time in order to identify patterns used to forecast in time and space by extrapolation. This has been proved to be effective for some applications (*e.g.* wind farm power production forecast), but requires of long series of wind observations that, for the application considered here, do not exist. On the other hand, dynamical

or physical downscaling (e.g. Castro et al., 2014) couples different nested models so that the outer mesoscale model furnishes boundary conditions to the inner microscale solver at each model time integration step. The inner microscale model can range in complexity from simpler mass-consistent diagnostic models to a CFD solver. This option is clearly more attractive but, apart from the higher computational cost, still presents some challenges related to not well-resolved inconsistencies in the physics of models across scales. On top of this, a pure dynamical downscaling approach becomes computationally prohibitive

for hazard assessment purposes, where climatically representative wind series have to be considered (note that this implies thousands of coupled simulations in order to statistically cover all meteorological situations with its associated probability). Given these constrains, we adopt an intermediate Physical-Statistical strategy based on transfer functions, a concept inspired on methodologies used for micro-scale wind resource assessment over regional scales (e.g. Sanz Rodrigo et al., 2010).

The idea behind transfer functions is simple. Given a mesoscale wind field (in our case WRF-ARW at 1 or 3 km grid resolution,

see Section 4.1) and a set of microscale wind fields, each characterized by a reference wind direction $\phi$ and intensity (in our case 24 ALYA-CFDWind runs at 50 m grid resolution and 15° geostrophic wind binning, see Section 4.2), the transfer functions determine a new (downscaled) wind field as:

$$u_{down} = f \times u_{WRF \rightarrow CFD} = \frac{u^{\theta}_{CFD}}{<u^{\theta}_{CFD}>_R} \times u_{WRF \rightarrow CFD} \qquad (10)$$

where $u_{down}(x,y,z,t)$ is the resulting downscaled wind velocity, $f = f(\theta,t)$ is the point-dependent transfer function at time

$t$, $u_{WRF \rightarrow CFD}$ is the WRF wind velocity interpolated to the (finer) microscale mesh, $u^{\theta}_{CFD}$ is the ALYA-CDFWind velocity for a reference wind direction $\theta$, and $<u^{\theta}_{CFD}>_R$ is the CDFWind velocity for the same direction $\theta$ averaged over a radius of influence $R$ (of size similar to that of the WRF cells). Note that, by construction, at each point, the CFD field $u^{\theta}_{CFD}$ averaged over its circumference of influence $R$ equals the WRF velocity interpolated at that point. In other words, the resulting downscaled wind field coincides on average (over a WRF cell) with that of the mesoscale model but, at the same time, it has all

the local wind fluctuations around this mean (caused by the microscale topographic forcing) which can not be captured by the (coarser) mesoscale grid. Moreover, the methodology is thought to preserve flow characteristics due to the thermal stratification present in the WRF model.

A central point in this hybrid downscaling methodology is how to determine the reference wind direction $\theta$ and then how to link it consistently with the mesoscale flow. For this, we adopt the following strategy to obtain downscaled 2D fields at given

heights $z_t$ above the terrain:

1. The first step consists in extracting from the microscale computational domain (ALYA-CFDWind) a series of 2D plains $\Omega_{2D}$ at user-defined heights above the terrain (*e.g.* $z_t =$2, 10, and 50 m). Each of these $\Omega_{2D}$ planes is then decomposed in a series of structured patches or segments $S_{ij}$, allowing for some overlap at the borders of the sub-domains (*i.e.*





$\cup S_{ij} = \Omega_{2D}; \cap S_{ij} \neq \emptyset$). In particular, we consider here one squared patch around each WRF mass-point with sizes $2dx_{WRF} \times 2dy_{WRF}$, being $dx_{WRF} \times dy_{WRF}$ the WRF cell area (see Figure 9).

2. For each sub-domain $S_{ij}$, the reference wind direction $\theta$ at time $t$ is computed as the averaged WRF velocity over the segment. Note that, for small computational domains or flat terrains, little variations in $\theta$ are expected across different segments $S_{ij}$. However, over large areas or in very complex terrains, synoptic-scale effects may result on variations of $\theta$ at different segments.

3. For each sub-domain $S_{ij}$ and time $t$, determine the microscale (ALYA-CFDWind) reference solution $u_{CFD}^{\theta}$ performing a linear interpolation between the two reference runs $\phi_1$ and $\phi_2$ that limit the bin direction containing $\theta$ (*i.e.* $\theta \in (\phi_1, \phi_2)$). For example, if for a given segment and time one has $\theta = 5°$, then the bin-bounding solutions $u_{CFD}^{0}$ and $u_{CFD}^{15}$ (*i.e.* precomputed solutions for $\theta = 0°$ and $\theta = 15°$ respectively) are combined to obtain $u_{CFD}^{5}$.

4. Perform a smoothing operation in the overlap regions between neighboring sub-domains $S_{ij}$. This is necessary because different values of $\theta$ in adjacent segments can result on discontinuous values of $u_{CFD}^{\theta}$ across segments. In particular, we consider a simple weighted interpolation in the regions with overlap.

5. Finally, apply the transfer functions using eq. (10) to scale the microscale wind modulus and obtain $u_{down}$.

Figure 10 compares the 3 km resolution WRF results at 10 m above the terrain with the downscaled field at the area of interest around the Lake Nyos. This Figure highlights the local information added by the microscale model over mountainous areas and valleys, where WRF shows a smoother behavior. Unfortunately, no surface wind data existed at that time to validate these results. However, some consistence with reports exists. For example, a strong microscale wind rotation from NE to SW (wind origin direction) is clearly visible along the Nyos valley from 18:00 to 21:00 UTC, in agreement with cloud dispersal reports indicating two differentiated dispersal phases. In order to illustrate this phenomenon, Figure 11 plots time series of 10-m wind velocity and direction from 16:00 to 24:00 UTC at two locations of special interest, the lake itself and a point at the bottom of the Nyos valley (see Fig. 3). Note how near surface wind direction changes sharply at both locations between 17:30 (wind from NE) and 19:30 (wind from SW) UTC. A WRF model error phase of about 3 hours seems to exist because dispersal reports suggest such a strong wind veering occurring after 22:00 LT (21:00 UTC). In any case, these wind field variations strongly indicate the need for including time-dependent heterogeneous winds for the gas dispersal simulations. Finally, it is worth looking at the atmospheric stability during the time of the event since this parameter also favors the accumulation of dense gas at depressions. Figure 12 plots the static stability versus time, clearly reflecting a unstable to stable transition related to the diurnal cycle occurring at around 16:00 UTC.

## 5   High-resolution dispersal modeling results

Once the high-resolution winds (*i.e.* WRF-ARW winds downscaled with ALYA-CFDWind using transfer functions) have been obtained for the period and area of interest, 10-m wind values every 20 minutes (*i.e.* 3 times hourly) were given to



the TWODEE-2.1 model together with the 30-m resolution DEM and the definition of the source term to simulate the $CO_2$ cloud dispersal. Our simulations assume the eruption started at 17:30 UTC (18:30 LT), *i.e.* 2-3h advanced with respect to eyewitness reports. This source term shift in the dispersal simulations was necessary because of the WRF model phase error discussed in Section 4.3. On the other hand, given the large uncertainties in the source term concerning eruption duration,

intensity and evolution of $CO_2$ mass flow rate with time, we performed a source term characterization considering different sets of simulations, each set with different source term characteristics (see Table 4):

- Eruptions lasting 3h with a constant $CO_2$ emission but varying the source intensity. This is similar to the scenarios considered by Costa and Chiodini (2015), who ranged the source intensity between $1.4 \times 10^5$ kg m$^{-2}$d$^{-1}$ (based on Kanari (1989)) and $4.3 \times 10^5$ kg m$^{-2}$d$^{-1}$ (based on Evans et al. (1994)) (Group 1 in Table 4);

- Eruptions lasting 3h with a $CO_2$ emission decreasing linearly and considering different slopes and source intensities (Group 2);

- Eruptions lasting 3h with a $CO_2$ emission increasing linearly and considering different slopes and source intensities (Group 3);

- Eruptions lasting 3h with a $CO_2$ emission decreasing exponentially considering different decay rates and source inten-
sities (Group 4);

- Eruptions lasting 3h with a $CO_2$ emission increasing exponentially considering different growth rates and source inten-
sities (Group 5);

- Eruptions with an initial phase with low constant $CO_2$ emission followed by a burst and exponential decay considering different durations for each phase, decay rates and source intensities (Groups 6 and 7). In addition, eruption durations
for these groups were varied between 2 and 4h as in Costa and Chiodini (2015).

Results for all these simulations (a total of 62 source term scenarios) were validated on a point-by-point basis comparing the TWODEE-2.1 predicted probability of fatalities with the actual percentage of fatalities reported at 53 locations (Table 1). When defining a metric for the skill score of a simulation, it is important to consider if observations are well-distributed across the possible range of values or not. In our case, for example, 29 observations of 53 (*i.e.* 55%) and 8 observations of 53 (*i.e.* 15%)
have 0 and 100% of observed fatalities respectively. It means that around 70% of the values are at the tails of the distribution and, consequently, the evaluation of scores without binning would lead to biassed skills, favoring those scenarios in which the source intensity (emitted mass) is largely under/over-predicted. In order to prevent this, we adopt binning strategy and evaluate scores first for each discrete bins and then globally by averaging over all bins. Ideally, the number of bins should scale as $\sqrt{n}$ so that, in our case ($n = 53$), around 7 bins would be recommendable. However, in order to avoid having bins with little or even
no observations, we considered the following 5-class binning for the percentage of fatalities: the first bin 0-5% (no significant mortality), the second bin 5-35%, the third 35-65%, the fourth 65-95%, and the fifth 95-100% (total mortality). This allows to




compute the Pearson cumulative test statistic $\chi^2$ as:

$$\chi^2 = \sum_{i=1}^{n_b} \frac{(O_i - M_i)^2}{M_i} \tag{11}$$

where $n_b = 5$ is the number of bins, $O_i$ is the number of observations (localities) within the $i-th$ bin, and $M_i$ is the number of model localities laying in the same bin. On the other hand, we also compute the total Mean Absolute Error (MAE) as:

$$MAE = \frac{1}{n_b} \sum_{i=1}^{n_b} (MAE)_i \tag{12}$$

with

$$(MAE)_i = \frac{1}{m} \sum_{j=1}^{m} |PO_j - PM_j| \tag{13}$$

where $(MAE)_i$ is the bin absolute error, $m$ the number of observations (localities) in the $j-th$ bin, $PO$ is the observed percentage of fatalities, and $PM$ is the modeled probability at the same locality.

As a best-fit criterion we considered the lowest value of $\chi^2$ but trying also to minimize MAE. We found that the higher-score runs belong to the same group of source term runs: a source with two phases; an initial phase with a constant $CO_2$ emission followed by a second phase with exponential source strength decay (group 6). This is a consistent result. Table 5 and Figure 13 summarize the characteristics of the source term for the highest ranked run. The total $CO_2$ emitted mass is 0.86 Tg, released during 3 h. This value is in good agreement with previous independent estimations, ranging from 0.29 Tg (Evans et al., 1994) to 1.95 Tg (Tuttle et al., 1987). On the other hand, the values of $\chi^2$ and MAE are 4.98 and 34.7% respectively, giving a very good fit across all bins (see Figure 14). For comparison, the scenario-II in Costa and Chiodini (2015) (see Figure 4) gives a $\chi^2$ of 7.61 and a MAE of 36.6%, substantially higher even if the characteristics of the source term are similar to our best-fit runs both in terms of duration (4 h) and total mass (1.33 Tg). These differences can be explained because of the high-resolution time-dependent winds, which are able to capture the wind veering leading to the differentiated cloud dispersal branches (see Figure 13). This can also be observed by looking at the evolution of concentration with time at different locations. As observed in Figure 15, the TWODEE-2.1 simulations can reproduce a first W-NW branch affecting the bottom valleys of Cha and Fang (location L34) followed by a gas cloud direction shift towards NE affecting the Subum village (location L36). According to simulations, this occurred around 2 hours after the eruption onset (*i.e.* 20:30 LT). Considering the 2-3 h shift necessary to correct the meteorological (WRF) phase error, this is in excellent agreement with eyewitness reports.

## 6 Summary and Discussion

We have developed a high-resolution numerical model for $CO_2$ atmospheric dispersal in complex terrains by introducing a methodology for time-dependent microscale wind characterization based on transfer functions that couple a mesoscale Numerical Weather Prediction Model with a microscale Computational Fluid Dynamics model for the atmospheric boundary layer. The model was applied to reconstruct the source conditions and the catastrophic $CO_2$ dispersal at Lake Nyos in 1986. Simulation results were compared with the observed fatalities through a novel probabilistic approach. We found that the new model





with high-resolution time-dependent winds shows better agreement with the observations compared with previous simulations, indicating that the model is capable of performing gas dispersal hazard assessment on very complex terrains in the future.

The optimal runs (higher scores for percentage of fatalities) shared same source evolution pattern: an initial phase with a low constant $CO_2$ emission and a second phase with a burst followed by exponential decay in $CO_2$ mass flux. This suggests additional information on physical processes at Lake Nyos during the 1986 limnic eruption. The scenario is compatible with a gas eruption that started locally in some region of the lake continuously emitting a relatively low amount of $CO_2$ near the equilibrium state and then, a perturbation, likely due to the gas ascent itself or to other mechanism(s), grew destabilizing the whole lake and triggering the exponential phase by overturn of the stratified layer in the lake in a supersaturation state. One simple model explaining the exponential decay during the second phase is that the driving force for $CO_2$ emission is an overpressure in the lake, such as caused by $CO_2$-supersaturation, and the temporal change in the overpressure is controlled by the emitted $CO_2$ flux, like it occurs in a volatile supersatured magma chamber. This model can be simply formulated as (e.g. Huppert and Woods, 2002):

$$Q = C_1 P \tag{14}$$

and

$$\frac{dP}{dt} = -C_2 Q \tag{15}$$

where $Q$ is the emitted $CO_2$ flux, $P$ is the overpressure, and $C_1$ and $C_2$ are two constants. These equations lead to the solution of exponential decay form for $Q$:

$$Q(t) = Q_0 \exp(-C_1 C_2 t) \tag{16}$$

where $Q_0$ is the initial value of $Q$. Although the precise mechanism in the lake during the limnic eruption is still under discussion, we can infer from the above model that the second phase is simply explained by a relaxation process of supersaturation state in the lake. Although further studies for the dynamics of lake water are necessary for reproducing the physical process of the limnic eruption in detail, the features of the $CO_2$ emission obtained in this paper may provide strong constraints on physical modeling in the lake.

*Author contributions.* AF has implemented the improvements to the TWODEE code and written the bulk of the manuscript with inputs from all authors. JB and AF have performed the wind field characterization. TK has configured and performed TWODEE simulations. AC has characterized the scenarios, defined the strategy for the impact criterion, and contributed to the text. All authors reviewed the manuscript.

*Acknowledgements.* JB has been partially funded by the Industrial Doctorate Program of the Catalan Government (eco/2497/2013).





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



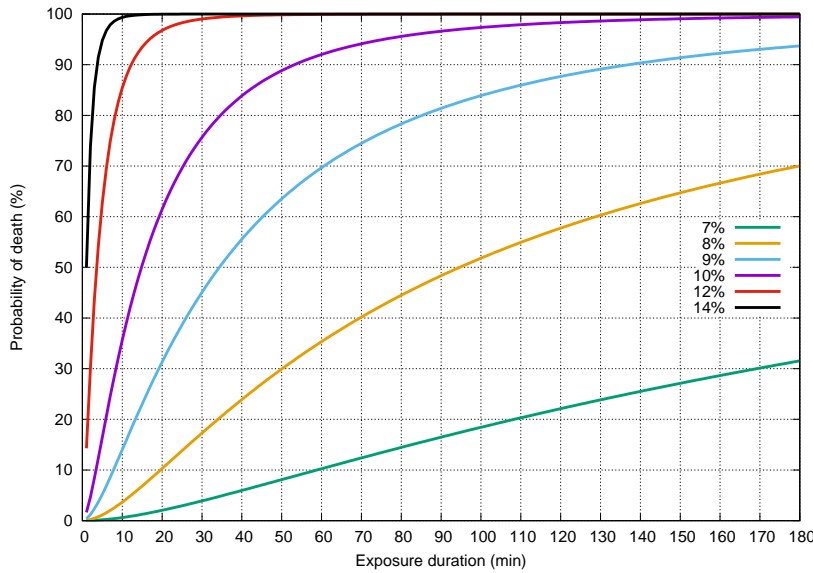

**Figure 1.** Percentage of fatalities (probability of death in %) depending on exposure duration (in min) according to eq. (7). Values are shown for different values of $CO_2$ concentration ranging from 7% to 14% vol.

Skamarock, W. C., Klemp, J. B., Dudhia, J., Gill, D. O., Barker, M., Duda, K. G., Huang, X. Y., Wang, W., and Powers, J. G.: A description of the Advanced Research WRF Version 3, Tech. rep., National Center for Atmospheric Research, 2008.

Tuttle, M., Clark, M., Compton, H., Devine, J., Evans, W., Humphrey, A., Kling, G., Koenigsberg, E., Lockwood, J., and Wagner, G.: The 21 August (1986) Lake Nyos gas disaster, Cameroon, Tech. rep., US Geol Surv Open-File Rept, 1987.

5 Vázquez, M., Houzeaux, G., Koric, S., Artigues, A., Aguado-Sierra, J., Arias, R., Mira, D., Calmet, H., Cucchietti, F., Owen, H., Taha, A., Burness, E. D., Cela, J., and Valero, M.: Alya: Multiphysics engineering simulation toward exascale, Journal of Computational Science, 14, 15–27, doi:http://dx.doi.org/10.1016/j.jocs.2015.12.007, 2016.

Zhang, Y.: Dynamics of CO2-driven lake eruptions, Nature, 379, 57–59, doi:10.1038/379057a0, http://dx.doi.org/10.1038/379057a0, 1996.





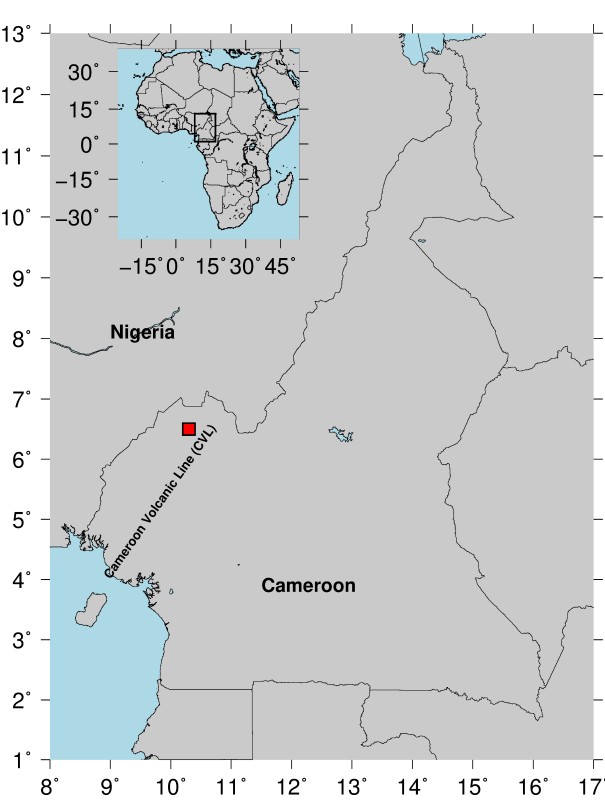

**Figure 2.** Map of Cameroon showing the lake Nyos area (red square).



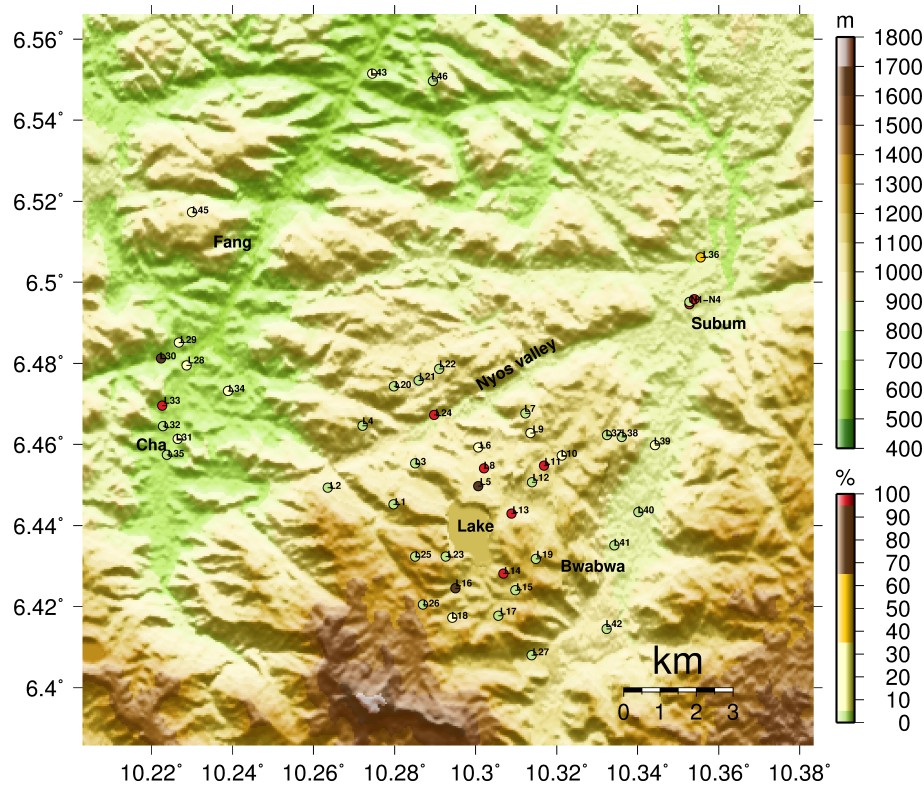

**Figure 3.** Topographic map of the area around Lake Nyos showing elevation contours in m a.s.l. The locations listed in Table 1 are shown by circles colored according the percentage of fatalities (in %) reported by Le Guern et al. (1992).

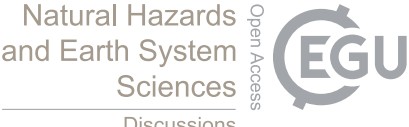



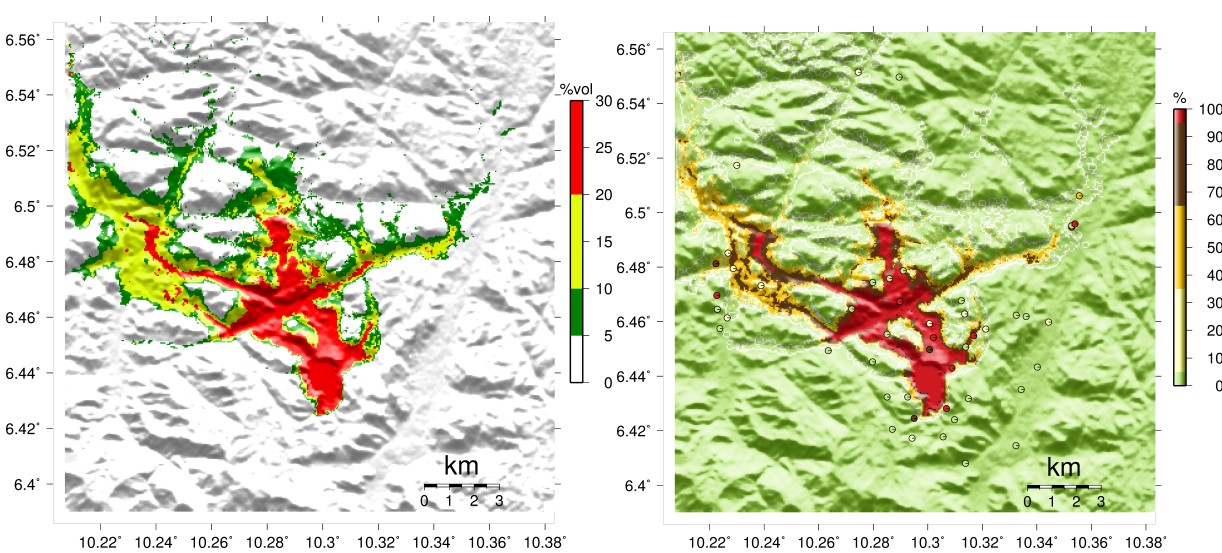

**Figure 4.** Previous modeling results from Costa and Chiodini (2015) scenario-II (4h of gas emission assuming a constant mass flux of $1.4 \times 10^5$ kgm$^{-2}$d$^{-1}$ from a diffuse source of $235 \times 235$m$^2$). Left: maximum $CO_2$ concentration (%vol.) achieved at 1 m height. Right: percentage of fatalities predicted by the model applying eq. (7) at 1 m height. Points show the actual reported percentages at locations using the same color scale.

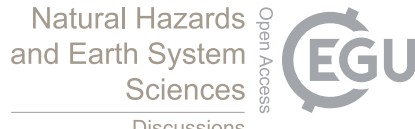

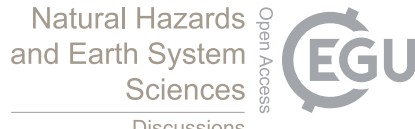

**Figure 5.** WRF-ARW domains for the Nyos run. The model configuration consists of one parent domain (d01) at 81 km horizontal resolution and 4 nests (d02 to d05) at 27, 9,3, and 1 km resolution centered at the Lake Nyos (red triangle, not on scale). Color contours indicate the WRF-ARW model topography at each resolution.




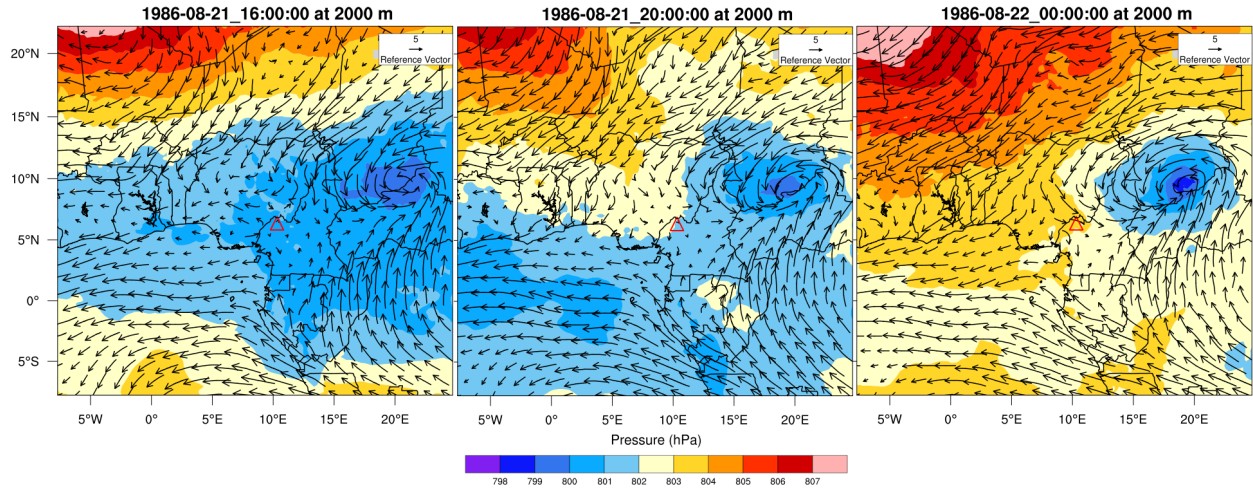

**Figure 6.** Synoptic meteorological situation according to the WRF-ARW simulation showing pressure contours (hPa) and wind vectors (m/s) for domain d02 (first nest, 27 km resolution domain) at 2000 m height above sea level. Results for 21 Aug 1986 at 16:00 (left), 20:00 (center) and 24:00 (right) UTC. The red triangle shows the location of the Nyos lake. For clarity, only few model wind vectors are shown.

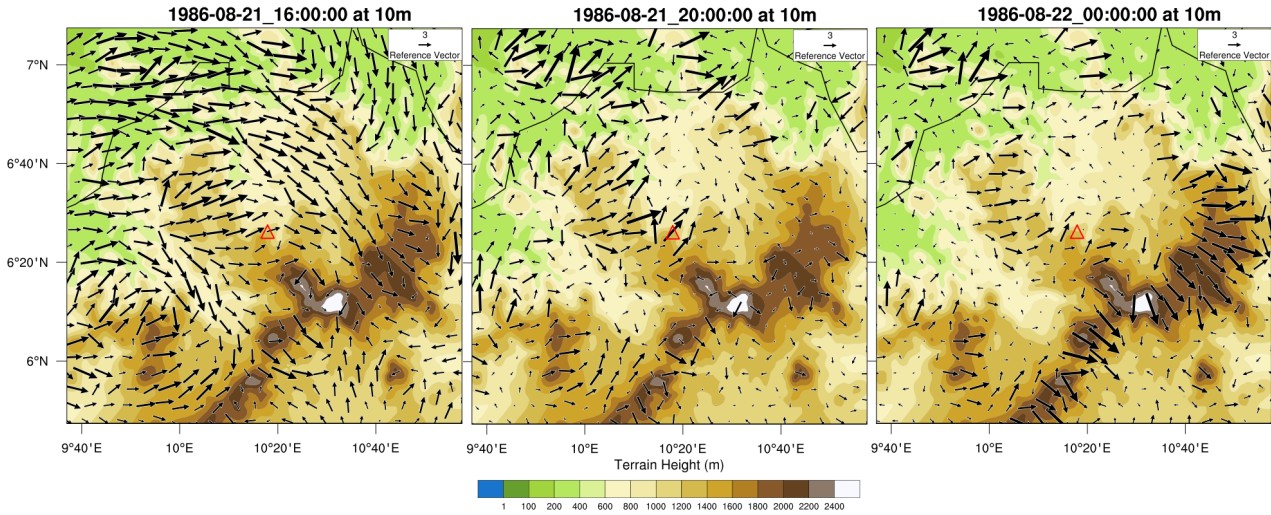

**Figure 7.** Local meteorological situation according to the WRF-ARW simulation showing terrain height contours (m a.s.l.) and wind vectors (m/s) for domain d05 (last nest, 1 km resolution domain) at 10 m above the surface. Results for 21 Aug 1986 at 16:00 (left), 20:00 (center) and 24:00 (right) UTC. The red triangle shows the location of the Nyos lake. For clarity, only few model wind vectors are shown.



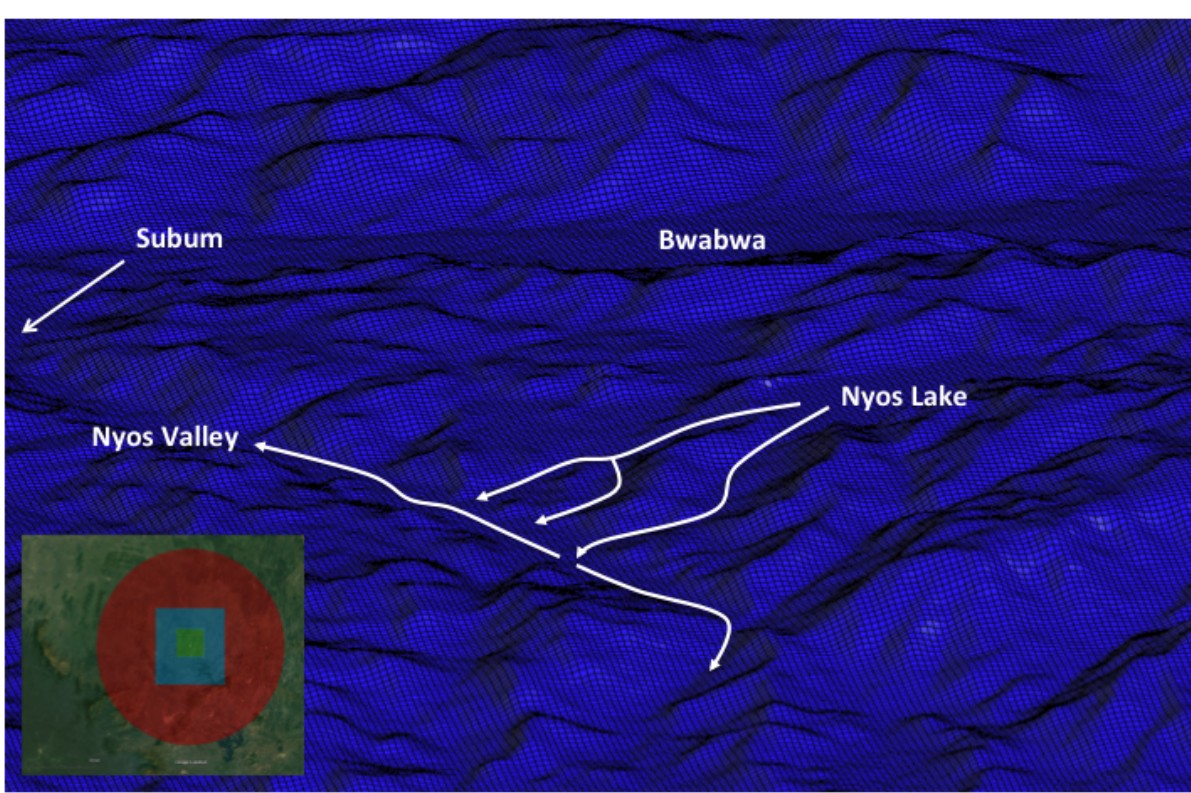

**Figure 8.** Detail of the ALYA-CFDWind computational mesh (50m horizontal resolution) around the Nyos lake. The bottom-left inset shows the extent of the computational domain composed of 3 differentiated zones, flat buffer (red), transition (pale blue), and inner domain (green) at 50m resolution containing the detailed terrain information. The arrows indicate the approximate gas flow path according to observations.





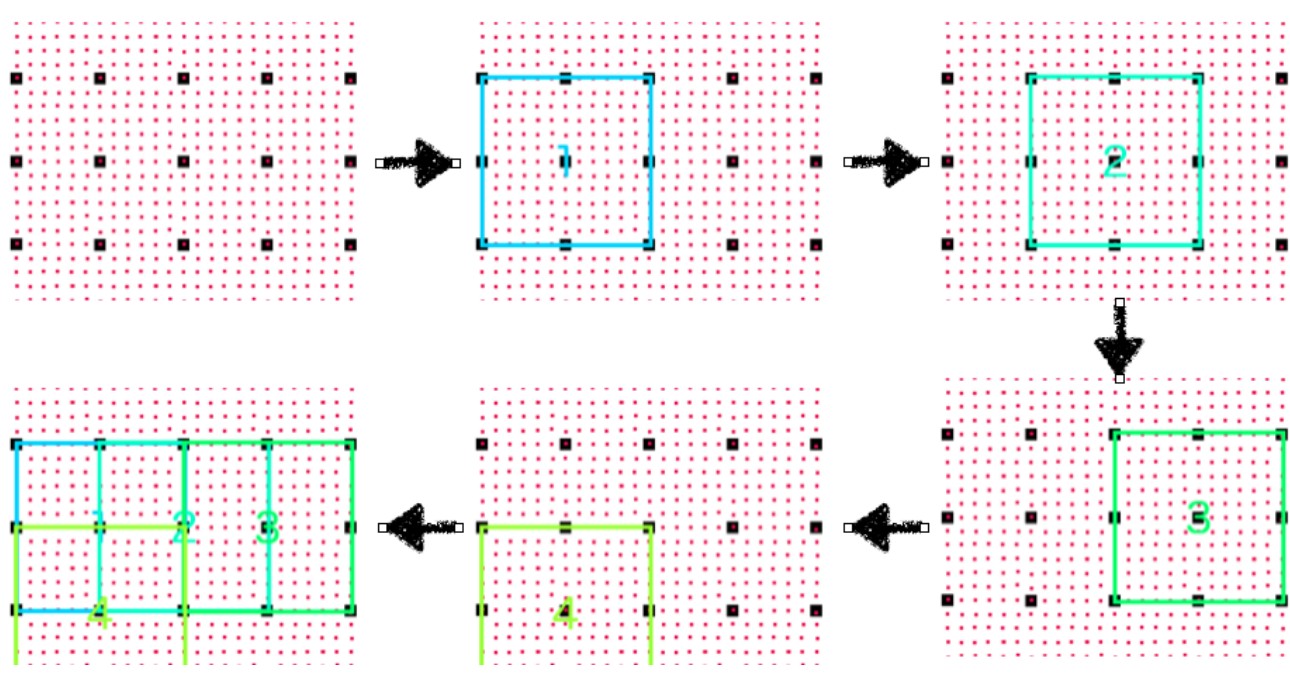

**Figure 9.** Segmentation strategy adopted for downscaling on a 2D plane $\Omega_{2D}$. A patch or segment $S_{ij}$ is defined for each WRF-ARW grid mass-point (black squares) containing many ALYA-CDFWind grid points (red small squares). The last plot shows the overlapped region.



**Figure 10.** Wind vectors at 10 m above terrain as given by WRF-ARW at 3 km resolution (left column) and the downscaling (right column) for 21 Aug 1986 at 18:00 (top), 19:00 (middle) and 21:00 (bottom) UTC. For visualization and point-to-point comparison pruposes, WRF results have been interpolated to the CFD mesh and, when necessary, extrapolated below its lower level. The red triangle shows the location of the Lake Nyos.



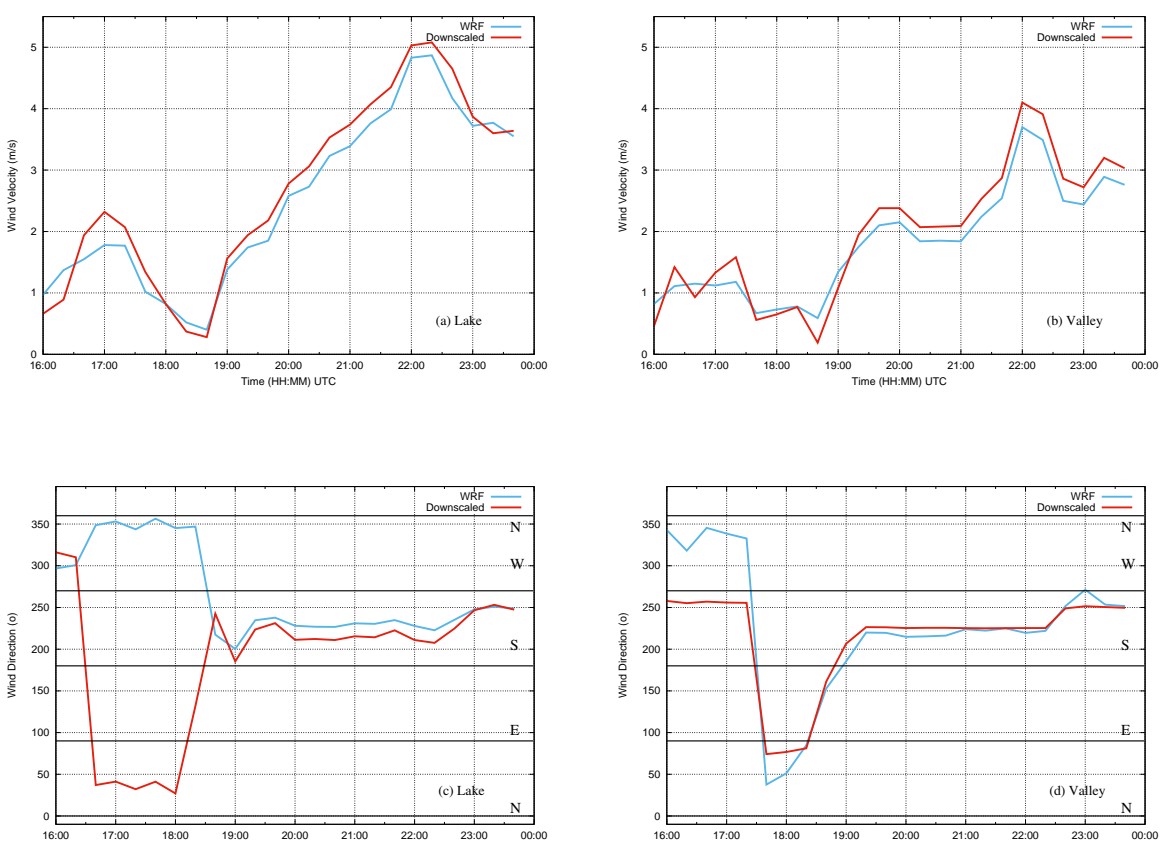

**Figure 11.** Time series of 10-m wind speed and direction at two points located at the Nyos lake (a,c) and at the bottom of the Nyos valley (b,d). Results from WRF simulations at 1 km resolution (blue lines) and downscaling using ALYA-CFDWind at 50 m (red lines). Wind direction criterion is that of the coming direction, *i.e.* $0^o$ indicates wind coming from the N, $90^o$ coming from the E, etc. Note the sudden short-lived wind veering from NE to SW starting at around 18:00 UTC.




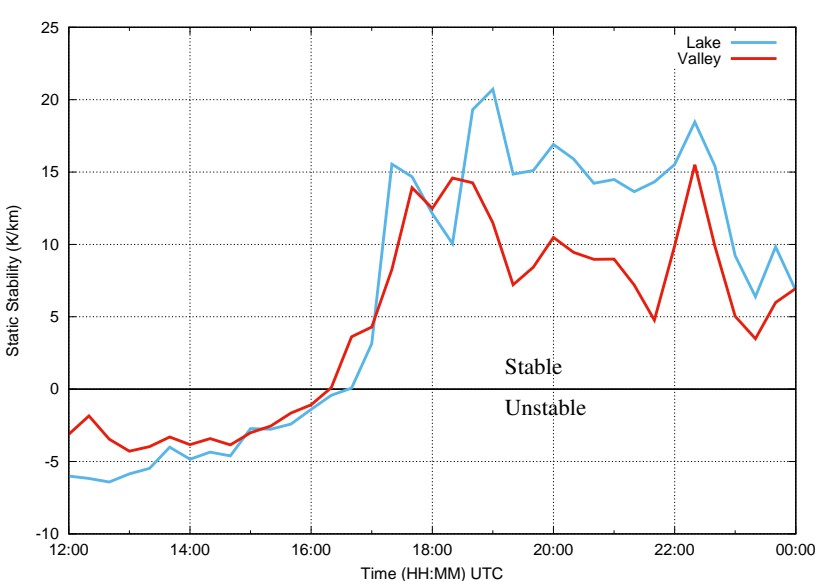

**Figure 12.** Static stability (gradient of potential temperature in $^{\circ}\mathrm{K\,km}^{-1}$) versus time at the same two points than in Fig. 11 (lake in blue and valley in red) according to WRF simulations at 1 km resolution. The potential temperature gradient has been computed using WRF $\sigma$-levels 1 and 6, at roughly 12 and 100 m above terrain respectively. Positive values indicate stable atmosphere. Note the transition from unstable to stable stratification occurring after 16:00 UTC.




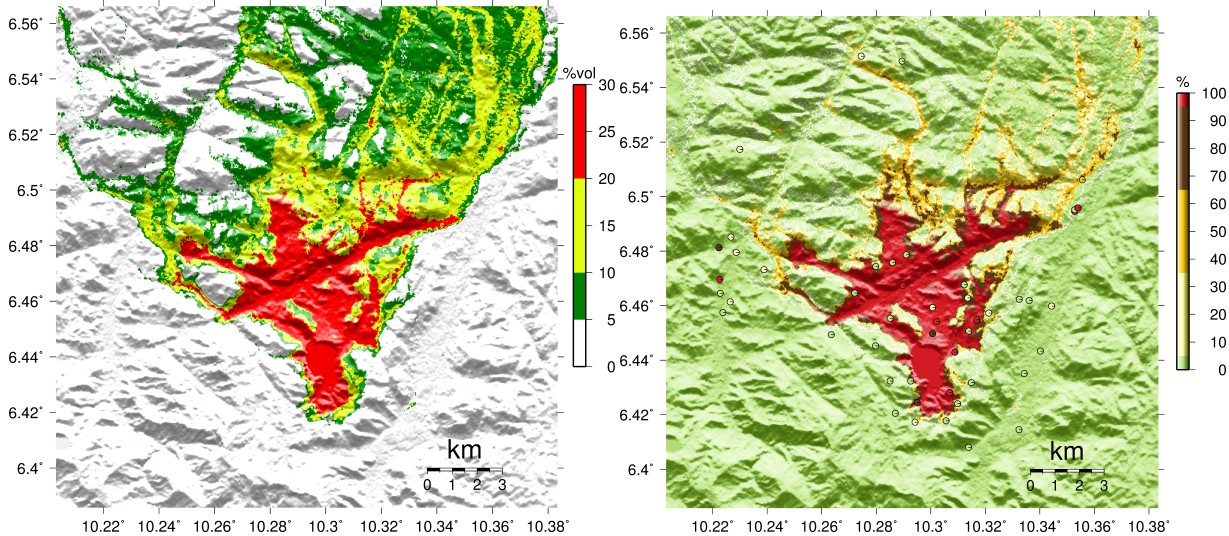

**Figure 13.** Best fit simulation results. Left: maximum $CO_2$ concentration (%vol.) achieved at 1 m height. Right: percentage of fatalities predicted by the model applying eq. (7) at 1 m height. Points show the actual reported percentages at locations using the same color scale.

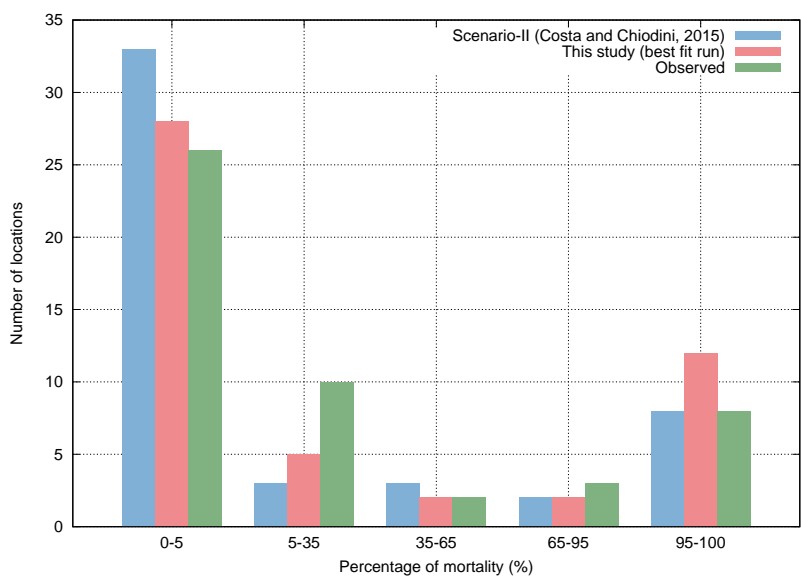

**Figure 14.** Histogram showing the fit between observations (green bars) and best-fit run (red bars) across the bins. Results from Costa and Chiodini (2015) are also shown for comparison (blue bars).





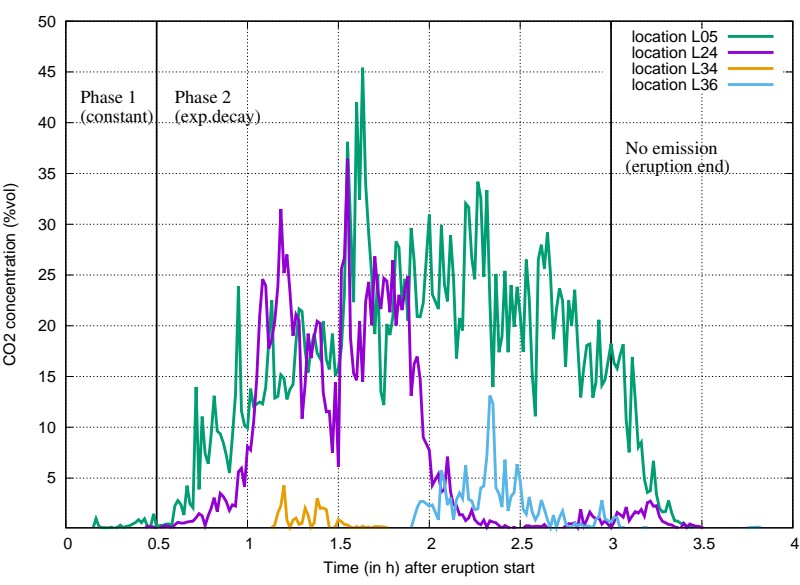

**Figure 15.** Evolution with time (in h) of $CO_2$ concentration (%vol.) at different locations for the best fit simulation. Results at 1 m height. Location L05 is near the Lake, L24 at the Nyos valley, L34 near Cha (W branch) and L36 near Subum (E branch). See Figure 3 for details.



**Table 1.** List of affected points with coordinates and percentage of observed fatalities. $(x, y)$ coordinates are in UTM zone 32N (datum WGS84). Points starting with an L have been digitalized from Le Guern et al. (1992); points starting with an N were obtained by the authors after interviewing survivors in situ. Comments as reported by Le Guern et al. (1992) in their Table 1.

| Point | x (m) | y (m) | lon (o) | lat (o) | Fatalities (%) | Comments |
|-------|-------|-------|---------|---------|----------------|----------|
| L1  | 641523 | 712595 | 10.280 | 6.445 | 0   | |
| L2  | 639726 | 713051 | 10.264 | 6.449 | 0   | |
| L3  | 642113 | 713722 | 10.285 | 6.455 | 0   | |
| L4  | 640665 | 714741 | 10.272 | 6.465 | 0   | |
| L5  | 643829 | 713105 | 10.301 | 6.450 | 90  | |
| L6  | 643829 | 714151 | 10.301 | 6.459 | 20  | |
| L7  | 645116 | 715089 | 10.312 | 6.468 | 0   | |
| L8  | 643989 | 713588 | 10.302 | 6.454 | 100 | |
| L9  | 645250 | 714553 | 10.314 | 6.463 | 15  | |
| L10 | 646108 | 713936 | 10.321 | 6.457 | 15  | |
| L11 | 645625 | 713668 | 10.317 | 6.455 | 100 | |
| L12 | 645303 | 713212 | 10.314 | 6.451 | 0   | |
| L13 | 644740 | 712354 | 10.309 | 6.443 | 100 | |
| L14 | 644526 | 710719 | 10.307 | 6.428 | 100 | A man ran uphill, survived, all cows died |
| L15 | 644847 | 710263 | 10.310 | 6.424 | 0   | |
| L16 | 643212 | 710316 | 10.295 | 6.425 | 80  | Nobody at the compound, 50 cows died, 15 survived |
| L17 | 644392 | 709566 | 10.306 | 6.418 | 0   | |
| L18 | 643131 | 709512 | 10.294 | 6.417 | 25  | |
| L19 | 645411 | 711121 | 10.315 | 6.432 | 0   | |
| L20 | 641523 | 715813 | 10.280 | 6.474 | 0   | |
| L21 | 642193 | 715974 | 10.286 | 6.476 | 0   | |
| L22 | 642756 | 716296 | 10.291 | 6.479 | 0   | |
| L23 | 642944 | 711174 | 10.293 | 6.432 | 0   | |
| L24 | 642622 | 715035 | 10.290 | 6.467 | 99  | Nyos village: 6 lived, 600 died |
| L25 | 642113 | 711174 | 10.285 | 6.432 | 0   | Upper Nyos; 150 survived, 0 died |
| L26 | 642327 | 709861 | 10.287 | 6.420 | 0   | |
| L27 | 645303 | 708493 | 10.314 | 6.408 | 0   | Several compounds, 40 survived, 0 died |
| L28 | 635865 | 716376 | 10.229 | 6.480 | 20  | 12 people survived |
| L29 | 635651 | 716993 | 10.227 | 6.485 | 30  | |
| L30 | 635168 | 716564 | 10.222 | 6.481 | 65  | |

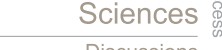
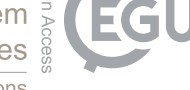
**Table 2.** cont.

| | | | | | | |
|---|---|---|---|---|---|---|
| L31 | 635624 | 714365 | 10.226 | 6.461 | 25 | 3 people lived |
| L32 | 635222 | 714714 | 10.223 | 6.464 | 0 | |
| L33 | 635195 | 715277 | 10.223 | 6.470 | 100 | 15 people died |
| L34 | 636991 | 715679 | 10.239 | 6.473 | 30 | Cha village: 130 survived, 58 died |
| L35 | 635329 | 713936 | 10.224 | 6.457 | 0 | |
| L36 | 649888 | 719352 | 10.356 | 6.506 | 50 | Subum village: 400 lived, 400 died |
| L37 | 647341 | 714499 | 10.332 | 6.462 | 0 | |
| L38 | 647743 | 714446 | 10.336 | 6.462 | 0 | |
| L39 | 648655 | 714231 | 10.344 | 6.460 | 20 | |
| L40 | 648199 | 712408 | 10.340 | 6.443 | 0 | |
| L41 | 647556 | 711496 | 10.334 | 6.435 | 0 | Mgombe: 160 lived |
| L42 | 647341 | 709217 | 10.332 | 6.415 | 0 | Upper Mgombe: 150 lived |
| L43 | 640906 | 724340 | 10.274 | 6.551 | 10 | |
| L44 | 642622 | 731794 | 10.290 | 6.619 | 5 | |
| L45 | 635999 | 720559 | 10.230 | 6.517 | 0 | |
| L46 | 642568 | 724152 | 10.289 | 6.550 | 0 | |
| L47 | 638305 | 727235 | 10.251 | 6.578 | 0 | |
| L48 | 641549 | 728844 | 10.280 | 6.592 | 0 | |
| L49 | 640557 | 729756 | 10.271 | 6.600 | 0 | |
| N1 | 649581 | 718074 | 10.353 | 6.495 | 100 | |
| N2 | 649581 | 718136 | 10.353 | 6.495 | 0 | |
| N3 | 649713 | 718210 | 10.354 | 6.496 | 50 | |
| N4 | 649719 | 718210 | 10.354 | 6.496 | 100 | |



**Table 3.** WRF-ARW model configuration and physical parameterizations used for the 1986 Lake Nyos simulations.

| WRF-ARW configuration | |
|---|---|
| Model version | 3.4.1 |
| Initial and Lateral BC's | NCEP/DOE Reanalysis-2 |
| Domains | 1 parent + 4 nests |
| Horizontal resolutions | 81 km (parent) and 27, 9, 3, and 1 km (nests) |
| Horizontal grid sizes | 110×110 (parent) and 136×127, 121×133, 181×166, and 151×151 (nests) |
| Vertical levels | 60 levels, with top at 70 hPa |
| Simulation length | 48 h (spin-up of 24 h) |
| Time step | 180 s (parent), ratios of 1/3 for nests |
| **Parametrization** | **Scheme** |
| Microphysics | WRF single-moment 6-class (WSM6) (Hong and Lim, 2006) |
| Cumulus | Modified Kain-Fritsch (disabled for 1 km nest) (Kain, 2004) |
| Surface Layer | MM5 Monin-Obukhov |
| Land Surface | Unified Noah Land Surface Model (LSM) (Chen and Dudhia, 2001) |
| Planet Boundary Layer | Mellor Yamada Janjic (MYJ) (Janjic, 1994) |
| Long-wave Radiation | Rapid Radiative Transfer Model (RRTM) (Mlawer et al., 1997) |
| Short-wave Radiation | Dudhia (Dudhia, 1989) |





**Table 4.** Source term settings used in the different groups of simulations. Table shows the considered ranges for eruption starting time, total $CO_2$ emitted mass, $CO_2$ mass flux, and constant of decay (or growth) for the exponential phase.

| Run group | Source type | Starting time (UTC) | Total mass (Tg) | Mass flux (given parameter) ($10^4$ kg m$^{-2}$d$^{-1}$) | Constant of decay (growth) for the exponential phase ($10^{-4}$ s$^{-1}$) |
|---|---|---|---|---|---|
| 1 | Constant (180 min) | 17:30 | $0.78 - 1.56$ | $10 - 20$ | - |
| 2 | Linear decrease (180 min) | 17:30 | $0.78 - 1.56$ | from $19 - 28$ to $1 - 2$ | - |
| 3 | Linear increase (180 min) | 17:30 | $0.78 - 1.56$ | from $1 - 2$ to $19 - 28$ | - |
| 4 | Exponential decrease (180 min) | 17:30 | $0.78 - 1.56$ | - | $2 - 10$ |
| 5 | Exponential increase (180 min) | 17:30 | $0.78 - 1.56$ | - | 5 |
| 6 | Constant (60 min) + Exponential decrease (120 min) | 17:30 | $0.16 - 2.34$ | $1 - 15$ (for constant phase) | $2 - 4$ |
| 7 | Constant (30 min) + Exponential decrease (90 to 210 min) | 16:30, 17:00, 17:30 | $0.16 - 2.34$ | $0.1 - 50$ (for constant phase) | $2 - 25$ |

**Table 5.** Source term characteristics of the best fit simulation showing total $CO_2$ emitted mass, mass flux variation, $\chi^2$, and Mean Absolute Error (MAE) considering the 5-class binning criterion.

| Duration (hour) | Total mass (Tg) | Mass flux ($10^4$ kg m$^{-2}$d$^{-1}$) | $\chi^2$ | MAE (%) |
|---|---|---|---|---|
| 3 | 0.86 | 4 for 30 min followed by a burst to 18 and 150 min exponential decay | 4.98 | 34.7 |