# Peer review of "High-resolution modelling of atmospheric dispersion of dense gas using TWODEE-2.1: application to the 1986 Lake Nyos limnic eruption"

_Natural Hazards and Earth System Sciences, 2016_

## Referee Comment (RC2) · Anonymous Referee #2 · 5 Apr 2017

The discussion paper submitted by Folch and co-authors presents an important improvement of the TWODEE-2.1 code, with the main impact being the methodology used to account with the microscale wind field characterization. In addition, the improvements give also the possibility to assess the impact in terms of human fatalities depending on the $CO_2$ concentrations and exposure time. Authors compared the results here obtained for Lake Nyos limnic eruption with results from a previous version of the TWODEE code (published by Costa and Chiodini, 2015), and check the fits between the observed fatalities and the results here presented. The good fits argue to justify the significant improvements of this new version of the code. In my opinion the article is quite interesting and a great improvement for the code that can be applied in

several degassing areas where CO2 constitutes a permanent hazard. In addition, the results now submitted for publication give some insights for the modeling of the 1986 Lake Nyos limnic eruption. Looking at figures 3, 4 and 13 there are some areas where fatalities occurred (for example L30 and L33) and that are not accounted by any of the models. It could be interesting to add some comment about this in the discussion (still some wind change that was not accounted or some limitation with the digital elevation model?). I have just few comments that can improve the paper, which are listed below: - Line 43 (page 1): I suggest to add the information that CO2 is denser than air at STP: "..., being denser than air at STP"; - Line 56 (page 1): comma is missing before "respectively" (other similar situations appear on the article); - Line 90 (page 2): remove one endpoint that is in excess; - Lines 5, 14, 17 and 21 (page 4): check and correct the number format. Periods indicate the decimal place in English, so the period should be replaced with comma (you want to separate groups of thousands); - Lines 5 to 22 (page 4): Authors use the thresholds mentioned by Costa and Chiodini (2015) to discuss the exposure limits for the CO2. References should be added for the symptoms and time of exposure mentioned for the 10%, for instance, since differences exist in the literature for the levels of CO2. As an example, some works mention that on the presence of 10% CO2 fainting can occur, so the 10-15 minutes mentioned by the authors in this article seem to be too long for exposure to these concentrations. In this paragraph authors should also mention the STEL for the CO2, which is 3%, value that is after used as the SLOT (it would be important to mention it before appearing as SLOT, as it is also defined as the STEL by OSHA, NIOSH and other international entities); Lines 47 and 51 (page 9): check the format of the CO2 (2 needs to be as subscript). Same comment for line 12 (page 10). Figures 4 and 13 - the dimension of the circles is too small (difficult to read). I suggest to increase the dimension of the circles.

---

## Author Comment (AC1) · 18 Apr 2017

We thank the anonymous reviewer #1 for his/her revision.

* Only a very minor, technical point about the quality of the figures 3, 4 and 13 where the coloured circles used to show the percentage of fatalities are too small and not easily readable and the colour scale in the legends should indicate "% of fatalities" and not just %.

Figures 3,4 and 13 have been changed according to the suggestion (enlarged circles and scale legend).

[Figure]

---

## Author Comment (AC2) · 18 Apr 2017

We thank the anonymous reviewer #2 for his/her revision.

o Looking at figures 3, 4 and 13 there are some areas where fatalities occurred (for example L30 and L33) and that are not accounted by any of the models. It could be interesting to add some comment about this in the discussion (still some wind change that was not accounted or some limitation with the digital elevation model?).

Minor comments and typos

o Line 43 (page 1): I suggest to add the information that $CO_2$ is denser than air at STP Done

o Line 90 (page 2): remove one endpoint that is in excess Done

o Lines 5, 14, 17 and 21 (page 4): check and correct the number format. Done

o Lines 5 to 22 (page 4): Authors use the thresholds mentioned by Costa and Chiodini (2015) to discuss the exposure limits for the $CO_2$. References should be added for the symptoms and time of exposure mentioned for the 10%, for instance, since differences exist in the literature for the levels of $CO_2$. As an example, some works mention that on the presence of 10% $CO_2$ fainting can occur, so the 10-15 minutes mentioned by the authors in this article seem to be too long for exposure to these concentrations. In this paragraph authors should also mention the STEL for the $CO_2$, which is 3%, value that is after used as the SLOT (it would be important to mention it before appearing as SLOT, as it is also defined as the STEL by OSHA, NIOSH and other international entities); We have enlarged this section and added references (Harper, 2011) as suggested.

o Lines 47 and 51 (page 9): check the format of the $CO_2$ (2 needs to be as subscript). Done

o Figures 4 and 13 - the dimension of the circles is too small (difficult to read). I suggest to increase the dimension of the circles. Done
* * *